# Observational Study on the Success Rate of Osseointegration: A Prospective Analysis of 15,483 Implants in a Public Health Setting

**Butruz Sarkis Simão, Jr. [1], Denis Damião Costa [2], Maria Cristina Teixeira Cangussu [3], Bruno Salles Sotto-Maior [4], Renan Lana Devita [5], Jorge José de Carvalho [6] and Igor da Silva Brum [7,*]**

1. Department of Implantology, São Leopoldo Mandic College, Campinas 13045-755, Brazil
2. Department of Dental Practice, Faculty of Dentistry, Federal University of Bahia, Salvador 40170-115, Brazil
3. Department of Social and Pediatric Dentistry, Faculty of Dentistry, Federal University of Bahia, Salvador 40170-115, Brazil
4. Faculty of Odontology, Federal University of Juiz de Fora, Juiz de Fora 36036-900, Brazil
5. Department of Orthodontics, Faculty of Odontology, University of Barcelona, 08193 Barcelona, Spain
6. Department of Histology and Embryology, State University of Rio de Janeiro, Rio de Janeiro 20950-000, Brazil
7. Department of Implantology, Faculty of Dentistry, State University of Rio de Janeiro, Rio de Janeiro 20950-000, Brazil
* Correspondence: igor_brum1@hotmail.com

**Abstract:** The osseointegration process between the host's bone tissue and the titanium implant is the key to success of implantology. The literature highlights the high success rate of osseointegrated implants, which is above 90%, and warns that some failures may occur, and every professional may face some inevitable failure. A longitudinal study was designed to evaluate the success rate of osseointegrated implants by taking into account the early failure of osseointegration. The study's population included a convenience sample of all patients attending in four municipalities in the state of Paraíba, Brazil, belonging to the Brazilian Unified Health System (SUS) and those who underwent implant placements between November 2015 and November 2018 and were followed-up until March 2020. Data were extracted from the National Registry of Health Establishments (CNES), a database that contains data on all Brazilian health institutions. Of the total placed implants, 1.88% failed before prosthetic rehabilitation, corresponding to a success rate of 98.12%. The success and failure rates by anatomical region were also evaluated, which revealed, respectively, the values for the anterior maxilla (95.52% and 4.48%), posterior maxilla (97.53% and 2.47%), anterior mandible (97.13% and 2.87%), and posterior mandible (98.90% and 1.10%). We conclude that the posterior mandible performed better than the other bone types and anatomical regions. The anterior region of the maxilla was the one that presented the worst performance. Moreover, when compared, the posterior maxilla performed similarly to the anterior mandible and better than the anterior maxilla.

**Keywords:** dental implant; osseointegration; longitudinal study; unified health system

## 1. Introduction

Dental implants are widely used in the rehabilitation treatments in private clinical implantology and are now used in Brazil in free treatments via the Public Health System (SUS). A milestone of implantology includes a public policy: the sanction of Ordinance No. 718/SAS on 20 December 2010 (which enabled the regulation for the placement of dental implants in the public health system) [1]. Currently, dental implants that are offered free of charge at the secondary level within the scope of oral health are tools that aim to change the image of a "country of toothless people", improving chewing, phonation, and aesthetics with the final goal of recovering the self-esteem of its users [2].

Dental implants represent an advantageous therapeutic option for replacing missing teeth, as they provide stable long-term support for dental prostheses subjected to chewing

loads [3]. The biological principle for the success of dental implants is osseointegration, a process that has been extensively investigated [3–5] and defined as the direct structural and functional connection between the living bone and the titanium surface of a load-bearing implant in the absence of an intermediate layer of connective tissue [6–11]. The use of titanium dental implants to support prosthetic rehabilitation has become a valuable and indispensable part of the treatment plan in modern and current Dentistry [12–14].

However, despite implant survival data that are close to 95–100% for total or partial rehabilitations, implants can fail [15–18]. Implant failures are classified as primary—when the organism does not reach osseointegration—or secondary—when the organism is unable to maintain the achieved osseointegration, resulting in a collapse of the repair [13]. Early failures occur before functional loading, and late failures occur after prosthetic placements [14].

The successful outcome of implant placement is mainly dependent on the interrelationship of the various components of an equation that includes the following: biocompatibility of the implant material; the macro- and microscopic nature of the implant's surface; the bone quality of the recipient bed; surgical technique; undisturbed repair phase; and loading conditions [19,20]. Local and systemic factors, such as low primary stability, surgical trauma and existing periodontal infection, may play an important role in hindering the normal bone repair process around the implants and, consequently, leading to early failure [21]. On the other hand, provisional overload and microbiologically induced peri-implant diseases are associated with late implant failure [22]. Even with high success rates (above 90%), some failures may occur, and every professional may face some inevitable failure at around 5% to 10%; when facing these situations, the dentist must be prepared to assess the probabilities of failure, possible complications, strategies to minimize failures, and the influence of bone density [23–28].

The Brazilian Unified Health System (SUS) is one of the largest and most complex public health systems in the world that aims to ensure comprehensive, universal, and free access to health services for the entire Brazilian population. Its creation provided access to health without discrimination. Comprehensive health care, and not only assistive care, has become a right for all Brazilians since their conception in life and it focuses on health with respect to improving the quality of life and aims at prevention and health promotion [2].

The objective of this study is to evaluate the success rate of osseointegrated implants placed in SUS patients attending four specific areas in Brazil and taking into account the early failure of osseointegration.

## 2. Materials and Methods

### 2.1. Study Design

This was a longitudinal study carried out in a convenience sample of all patients attending four municipal hospitals in the state of Paraíba, Brazil; these patients belonged to the Brazilian Unified Health System (SUS), underwent implant placement between November 2015 and November 2018, and were followed-up until March 2020. Data were extracted from the National Registry of Health Establishments (CNES), a database that contains data on all Brazilian health institutions, and the data corresponded to the four municipalities of the aforementioned state of Paraíba, Brazil (Table 1), where implantology services are guaranteed and free of charge via SUS in the Dental Specialties Centers (CEO) by Decree No. 718 / SAS of 20 December 2010 [1].

### 2.2. Participants and Study Procedures

Participants (*n* = 3690) underwent placement of dental implants by SUS in municipalities (Table 1) and returned for prosthetic rehabilitation 6 months after implant placement. To this end, they were clearly informed about the surgical implantation procedures and all volunteers provided written informed consent for the scientific use of their anonymous data. The study was conducted in accordance with ethical guidelines for research on human subjects. The study protocol was approved by the four Clinical Research Ethics Committees

of the municipalities where the patients were recruited and by Ethics Committee of Faculty of Dentistry São Leopoldo Mandic, Campinas-SP, Brazil.

**Table 1.** Participating municipalities.

| Municipality | Dental Specialist Center | National Registry of Health Establishments (CNES) |
|---|---|---|
| Brejo do Cruz—PB | Brejo do Cruz | 7481772 |
| Mogeiro—PB | Mogeiro | 7983565 |
| Pombal—PB | Pompal | 3990931 |
| Sumé—PB | Imaculada Conceição | 3738558 |

PB: Paraíba—Brazil.

The inclusion and exclusion criteria in addition to the characteristics of the study population are described in Tables 2 and 3.

**Table 2.** Inclusion and exclusion criteria.

| Inclusion | Exclusion |
|---|---|
| **Age** | **Age** |
| 18–75 years | <18 or >75 years |
| **Chronic disease** | **Chronic disease** |
| No | Yes |
| **Implant** | **Implant** |
| Systhex® | No Systhex® |
| ≥8.5 mm | <8.5 mm |
| Bone installed | Graft installed |
| ≥20 N·cm$^2$ | <20 N.cm$^2$ |
| **Prosthesis installation** | **Prosthesis installation** |
| 06 months | < or > 06 months |
| **Same surgical team** | **Different surgical team** |

**Table 3.** General characteristics of participants (*n* = 3690).

| Variables | N (%) |
|---|---|
| **Age** | |
| 18–75 years | 3690 (100%) |
| **Sex** | |
| Men | 1550 (42%) |
| Women | 2140 (58%) |
| **Color** | |
| White | 960 (26%) |
| Brown/Black | 2730 (74%) |

Regarding dental implants, only the Systhex® system implants, with an external hexagonal platform, different sizes and diameters (Systhex®, Curitiba-PR, Brazil) were included in this experiment, and they were placed by the same surgical team in the four municipalities, in different days or weeks, following the system company's placement instructions and protocol. Covers screws were placed on the implants and then the implants were kept submerged for 6 months, when they received the prosthesis.

Osseointegration was evaluated radiographically through periapical or panoramic radiographs, by the absence of a radiolucent image indicative of bone loss around the implant and, clinically, by the absence of implant mobility at 6 months, during the installation of the prosthesis over the implant. The surgical team consisted of 7 specialists in Implantology, trained with the Systhex® system and who were properly calibrated to install the implants, and subsequently assess the failure or success of osseointegration.

### 2.3. Outcome

The primary objective of the study was to assess the success rate of dental implants, including the early failures of the osseointegration process. Early failures were defined as those occurring before prosthetic rehabilitation and for 2 years after rehabilitation. The placed implants were separated by areas: anterior and posterior maxilla and anterior and posterior mandible. Osseointegration as the primary outcome of the study was assessed by a single examiner so that the inter-rater reliability was not applicable. The nature of this evaluation was to provide a basis for determining the "success rate" of this study (clinical: probing around the implant; implants with periodontal pockets smaller than 3 mm were considered healthy and radiological: periapical radiographs and CT scans were performed; images with a bone loss greater than 3 mm between the hexagon head and the apical part of the implant were considered unhealthy and were considered losses).

### 2.4. Statistical Analysis

Categorical data are expressed as frequencies and percentages. The percentages of success and failure corresponding to the different anatomical regions were compared using Pearson's chi-square ($\chi^2$) test for independent samples. Statistical significance was set at $p < 0.05$. The Statistical Packaged for Social Sciences (SPSS) version 17.0 for Windows was used for data analysis. The methodology and results were reviewed by a consulting independent statistician.

### 3. Results

During the study's period, a total of 15,483 dental implants (Systhex®) were placed in 3690 patients, with 1115 implants inserted in the anterior mandible, 8844 in the posterior mandible, 1272 in the anterior maxilla, and 4252 in the posterior maxilla. The overall success rate was 98.12%. Failure was recorded with an overall failure rate of 1.88% (Table 4 and Figure 1). Success rates range from 95.52% for implants placed in the anterior maxilla to 98.90% in those inserted in the posterior mandible. As shown in Table 3, the posterior mandible had the best performance and was statistically significant in relation to all other groups. The regions with similar results were the anterior mandible and posterior maxilla ($p = 0.470$), while the anterior maxilla had the worst result and was statistically significant when compared to all other groups, (Table 5).

**Table 4.** Success and failures rates by anatomical site of implant placement.

| Implant Region | Total Implants | Success | | Failure | |
|---|---|---|---|---|---|
| | | Number | Rate, % | Number | Rate, % |
| Mandible | | | | | |
| Anterior | 1115 | 1083 | 97.13 | 32 | 2.87 |
| Posterior | 8844 | 8747 | 98.90 | 97 | 1.10 |
| Maxilla | | | | | |
| Anterior | 1272 | 1215 | 95.52 | 57 | 4.48 |
| Posterior | 4252 | 4147 | 97.53 | 105 | 2.47 |

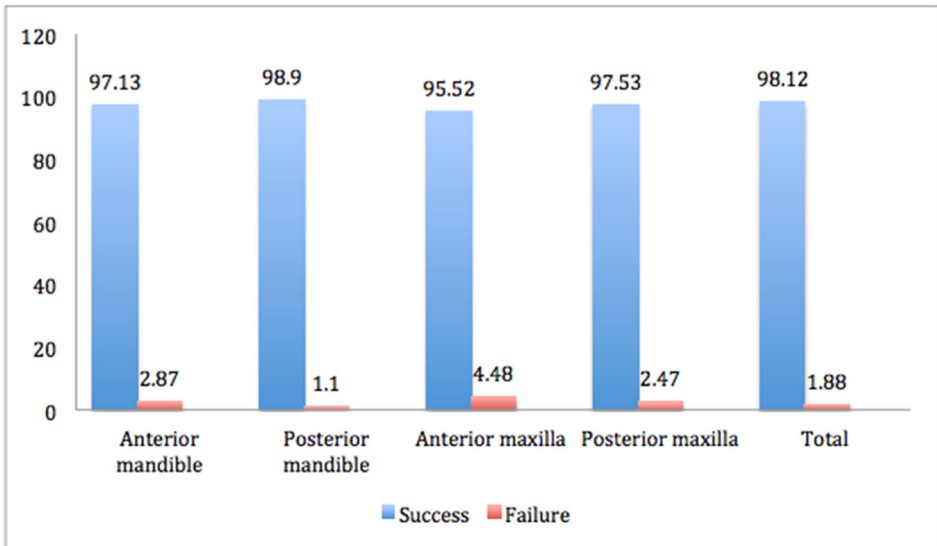

**Figure 1.** Early success and failure rates of dental implants according to anatomical locations.

**Table 5.** Differences in the results obtained from the anatomical site of the implant's placement.

| Study Groups | *p* Value * |
|---|---|
| Anterior mandible vs. posterior mandible | 0.001 |
| Anterior mandible vs. anterior maxilla | 0.035 |
| Anterior mandible vs. posterior maxilla | 0.470 |
| Posterior mandible vs. anterior maxilla | <0.001 |
| Posterior mandible vs. posterior maxilla | <0.001 |
| Anterior maxilla vs. posterior maxilla | 0.001 |

* Pearson's chi-square test.

## 4. Discussion

The present study adds evidence of the high rate of early success of dental implant surgery (success rate 98.12%), which is consistent with data that are previously reported by others [18,29]. A failure rate of 1.88% also indicates that there is still room for improvement [7,10,30]. Although it has been emphasized that factors related to bone quantity and quality, including bone mineral density, skeletal architecture, matrix properties, or the three-dimensional orientation of bone trabecula are important local factors affecting the success of dental implants [31–33], we found that the location where the implant was placed affected the outcome rather than the amount of bone.

The overall failure rate of dental implants varies between 5% and 10% [34], and the incidence of early failure (before prosthetic rehabilitation) is between 0.7% and 2.0% [35–40], which agrees with the rate of 1.88% found in the present study. In a study that comprised 4641 Brånemark dental implants, which were retrospectively followed from stage 1 surgery to the completion of prosthetic restorations, the rate of early failure was 1.5% [41]. In another study with 2670 patients who received 10,096 implants, early failures were reported in 1.74% of patients [42]. However, in a series of prospective studies started in the mid-1980s at the University of Toronto, a rate of failure of 4.2% before the insertion of prostheses was found [43].

On the other hand, predicting implant success is inherently a difficult challenge. Numerous studies have shown that implants osseointegrate more efficiently in relation to the classification of bone quality based on the relative proportion of compact cortical bones to spongy trabecular bones [44–49]. Dental implants placed in types I, II, and III bones were associated with successful outcomes [50,51], whereas prognosis appeared to be less favorable in type IV and is described as having a very thin cortex and low-density

trabeculae [52,53]. In the present study, however, the anatomic location influenced implant success. Implants in the posterior maxilla (type IV bone) showed a similar success rate than implants placed in the anterior mandible (type I) and superior to the anterior maxilla (types II and III). In fact, we found that the success rate of implants placed in the anterior maxilla was significantly lower compared to other anatomical regions. Other authors reported that the anterior maxilla is more critical for implant loss than other sites [54,55]. However, in a group of 126 patients with implant failure from a series of 3578 implant-treated subjects, most failures occurred before loading and were more frequent when the implant was installed in the posterior jaw (58.5%) [56].

Among the numerous tests available for diagnosing dental implant loss, we can cite three as the most relevant: The first involves probing the groove's depth formed around of the implant and the definitive prosthesis [57,58]; the second involves periapical radiography—with this radiographic technique, it is possible to accurately diagnose bone loss in two dimensions, which helps in the early treatment of peri-implantitis [59]; and finally, the third involves tomography—this exam allows the professional to quantify the level of bone crest that was lost and determine the best treatment to resolve the problem [60]. Corroborating what has been described in the literature, this study followed these three criteria as a basis for diagnosing early and late failures of dental implants.

The present findings should be interpreted by taking into account the limitations of the study, particularly the selection of patients as a non-randomized convenience sample based on subjects included in the Unified Health System and corresponding to four municipalities in Brazil. The fact that the success and failure rates of dental implants were assessed in the early period before prosthetic rehabilitation should also be considered; therefore, further studies with prolonged follow-up periods are needed to assess the mid- and long-term outcomes of implants following definitive prosthetic restorations.

## 5. Conclusions

In the present study of 15,483 implants placed in 3690 patients from the Unified Health System in Brazil, the early success rate was 98.12%. According to the anatomical site of implant placements, the most favorable results were obtained in the posterior mandible and the less favorable results were obtained in the anterior maxilla. Although the external validity of the present study is limited, studies carried out in the context of local health care systems are expected to improve knowledge and contribute to the integration of their applicability in everyday oral implantology practices.

**Author Contributions:** Conceptual-ization, B.S.S.J. and D.D.C.; methodology, M.C.T.C. and R.L.D.; validation, B.S.S.-M.; investigation, B.S.S.J. and D.D.C.; writing—original draft preparation, J.J.d.C. and I.d.S.B.; supervision, J.J.d.C. and I.d.S.B. All authors have read and agreed to the published version of the manuscript.

**Funding:** This research received no external funding.

**Institutional Review Board Statement:** The study was conducted in accordance with the Declaration of Helsinki and approved by the Institutional Review Board (or Ethics Committee) of the Regional society of education and health (protocol code 2.270.471 and date of approval: 9 December 2017). Written informed consent has been obtained from the patient(s) to publish this paper.

**Conflicts of Interest:** The authors declare no conflict of interest.

## Abbreviations

SUS      Unified Health System
CNES    National Register of Health Establishment

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
