# Peer review of "Observational Study on the Success Rate of Osseointegration: A Prospective Analysis of 15,483 Implants in a Public Health Setting"

_2673-8430, doi:10.3390/biomed2040033_

Round 1
Reviewer 1 Report
See the attached Word file

Author Response
Thank you for the excellent review.
I will do the English proofreading with the magazine's team

Reviewer 2 Report
I think there is nothing new in this study, the results are in accordance to already known information from the literature. The only advantage is the high number of implants placed.
Author Response
Dear reviewer
Thank you very much for the review,
The findings of this article is very important due to the great difficulty of including a high quality service in the public health service of brazil at zero cost to the patients.
Sincerely
Reviewer 3 Report
The manuscript titled: „Observational Study on the Success Rate of Osseointegration: A Prospective Analysis of 15,483 Implants in a Public Health Setting” evaluated the success rate of osseointegrated implants taking into account the early failure of osseointegration.
Introduction could be shortened. Lines 79-84 are somehow a repetition of what was written in lines 45-52; please revise, merge, shorten.
Put inclusion and exclusion criteria in a table.
Line 126-127: Correct the sentence
The study has a potential. However, little data was presented on the patient examination process. Explain, how exactly the success rate was determined in this study. Provide more details on osseointegration assessment and results. Provide more information on determinants (other than location) of success rate. All this could allow for more thorough analysis.
After improving Methods and Results, Discussion must be enhanced as well.
Author Response
Thank you very much for the excellent review
I have made all the considerations
I will also do the English proofreading with the magazine team

Reviewer 4 Report
Dear authors,
Congratulations on your work!
Author Response
Dear reviewer
Thank you very much for considering our manuscript.
I will do the English proofreading with the magazine's team
Sincerely
Round 2
Reviewer 2 Report
I still believe there aro no new findings in this manuscript, but I support the publication due to the high number of participants.
Author Response
Dear Review
Thank you very much for considering in the manuscript.
Sincerely.
Reviewer 3 Report
Thank you for revising the manuscript.
Thank you for putting inclusion and exclusion criteria in a table, but try to make it more clear/readable. What does “≠” mean? Also, since you put the information in Table, remove it from the text above. Exactly, how much time elapsed after implant placement: in the text "3 to 6 months" is mentioned, while table 1 mentions 6 months.
Still, little data was presented on the patient examination process. How many operators were involved in the examination process (you mentioned one), since the data come from four municipalities? Explain, how exactly the success rate was determined in this study. Provide more details on osseointegration assessment and results (bone loss measurements – any reference point in previous radiographs or CTs).
Provide more information on determinants (other than location) of success rate, e.g. age, gender, etc. all that is mentioned in group characteristics in Table 3.
Overall, the writing in the manuscript should be enhanced, English language and style.
Author Response
Dear review,,
Thank you very much for considering in the manuscript.
Sincerely
